# Design Strategies of Conductive Hydrogel for Biomedical Applications

**DOI:** 10.3390/molecules25225296

**Published:** 2020-11-13

**Authors:** Junpeng Xu, Yu-Liang Tsai, Shan-hui Hsu

**Affiliations:** 1Institute of Polymer Science and Engineering, National Taiwan University, No. 1, Sec. 4 Roosevelt Road, Taipei 10617, Taiwan; f07549033@ntu.edu.tw (J.X.); jimmy821212@gmail.com (Y.-L.T.); 2Institute of Cellular and System Medicine, National Health Research Institutes, No. 35 Keyan Road, Miaoli 35053, Taiwan

**Keywords:** conductive hydrogel, in situ polymerization, postmodification, composite, biosensors, drug delivery, tissue engineering

## Abstract

Conductive hydrogel, with electroconductive properties and high water content in a three-dimensional structure is prepared by incorporating conductive polymers, conductive nanoparticles, or other conductive elements, into hydrogel systems through various strategies. Conductive hydrogel has recently attracted extensive attention in the biomedical field. Using different conductivity strategies, conductive hydrogel can have adjustable physical and biochemical properties that suit different biomedical needs. The conductive hydrogel can serve as a scaffold with high swelling and stimulus responsiveness to support cell growth in vitro and to facilitate wound healing, drug delivery and tissue regeneration in vivo. Conductive hydrogel can also be used to detect biomolecules in the form of biosensors. In this review, we summarize the current design strategies of conductive hydrogel developed for applications in the biomedical field as well as the perspective approach for integration with biofabrication technologies.

## 1. Introduction

Hydrogels, as an important category of biomaterials, are composed by crosslinked polymer networks to form three-dimensional structures containing a high proportion of water [1,2]. Hydrogels could be excellent candidates to mimic the microstructures of natural cellular environments because of their similarity to the extracellular matrix. Many smart hydrogels (e.g., self-healing hydrogel and conductive hydrogel) have received remarkable attention for numerous biomedical applications such as tissue regeneration [3] and tissue adhesion [4,5]. Considering the increasing needs of biomaterials for healthcare [6], conductive hydrogels have recently started to attract attention from the scientific community because of their unique properties to provide promising applications in the biomedical field, such as implantable and wearable biosensors [7,8].

The conductive mechanisms of conductive polymers can be divided into ionic conductivity and electronic conductivity [9,10]. The electronic function of conductive polymers depends on the conjugated π system (i.e., π-π stacking) with the overlapped π orbitals throughout the polymer chain by polymerization. At the molecular level, the alternating patterns of single and double bonds in the presence of aromatic rings allow charge transfer [9,11]. Through aromatic stacking rings or the conjugated π system in conductive polymers, charge transfer can occur along chains (intra-chain transfer) or between conductive polymer chains within the hydrogel (inter-chain transfer) [11,12]. Thus, the electrical conductivity of conductive hydrogels relies on the charge transfer capability between or along different polymer chains [13,14].

Conductive hydrogels containing conductive polymers were initially fabricated as hybrid composites by combining conductive polymers with a preformed hydrogel, e.g., a polyacrylamide (PAM) hydrogel [15]. However, there are some shortcomings in these kinds of composite conductive hydrogels, such as uncertain polymerization residue, toxicity in vivo and nonbiodegradability, which restrict their applications in the biomedical field [13,16]. The new generation of conductive hydrogels is designed to use biofriendly polymers and different processing strategies to overcome the limitations, which can endow conductive hydrogels with biocompatibility and biodegradability. Conductive hydrogels for biomedical applications are typically composed of conductive polymers (e.g., polypyrrole [17] and polyaniline [18]), conductive particles (e.g., metals [19] and carbons [20]), and polysaccharides or proteins (e.g., gelatin/GelMA [21] and chitosan [18]). These conductive hydrogels have been widely utilized in electrosensitive tissues (e.g., cardiac muscles [22] and nerves [23]) because of the effective promotion of cell activity with or without electrical stimulation [24,25,26].

Conductive hydrogels are currently used in several biomedical fields (Figure 1), including biosensors [7,8], wound dressings [27,28], drug delivery [29,30] and tissue regeneration [31]. This review highlights the use of conductive hydrogels designed by various conductive strategies that can be applied in physiological environments. After a brief classification of the design strategies, the preparation and biomedical applications of conductive hydrogels are further considered.

## 2. Classification and Design

The rationale for integrating conductivity into a hydrogel network has been reviewed in previous literature [32,33]. The strategies for preparing the new generation of conductive hydrogels can be divided into three categories based on the source of conductivity and where the conductivity of hydrogel is provided by (i) in situ polymerization of conductive polymers, (ii) post polymerization of conductive polymers and (iii) composite strategies. The advantages and disadvantages of these different design strategies for preparing conductive hydrogels are briefly listed in Table 1 and the detailed contents are discussed below.

### 2.1. Conductivity Provided by In Situ Polymerization

This is a one-step preparation strategy where the conductive polymer monomer (e.g., pyrrole or aniline) is added or grafted into the matrix and the oxidant (e.g., ammonium persulfate (APS) or sodium persulfate (SPS)) is simultaneously added to complete the polymerization during the gelation process. This strategy is commonly used to prepare conductive hydrogels for biomedical applications. Through this strategy, the homogeneous in situ polymerization of conductive polymers can be triggered inside the hydrogel matrix in the presence of uniform oxidants. However, considering the poor solubility of most conductive polymers, it is necessary to modify the solubility of the conductive polymers to introduce them into the hydrogel matrix. Conductive polymer monomers are usually grafted to natural polymer, such as chitosan [18] and hyaluronic acid [28]. The monomers can also be made into nanoparticles [34] to achieve an acceptable level of solubility/dispersity in the hydrogel matrix. For example, Xu et al. [35] synthesized a self-healing conductive hydrogel though the Schiff base reaction. The in situ polymerized polypyrrole was prepared by mixing and stirring chitosan, pyrrole, and oxidizing these together in an ice bath before the gelation process. Zhao et al. [18] performed in situ polymerization of aniline by grafting monomers of aniline to quaternized chitosan with the addition of oxidized dextran as a dynamic Schiff crosslinker to form a conductive hydrogel network (Figure 2). The conductive hydrogel exhibited antibacterial property and a conductivity of 0.43 mS/cm. The conductive hydrogel synthesized by this strategy was proved to maintain cell viability and promote cell proliferation.

### 2.2. Conductivity Provided by Postpolymerization

This conductive strategy requires the preformed hydrogel matrix to be immersed in the monomer solution of the conductive polymer (or oxidant solution). Subsequently, the conductive hydrogel can form after immersion of the monomer-filled hydrogel into the oxidant solution (or immersion of the oxidant-filled hydrogel into the monomer solution) to finish the polymerization [36,37]. This process provides the possibility of giving the already formed hydrogel extra conductivity. Meanwhile, the post polymerization of conductive polymers, functioning as an alternative coating tool, offers high versatility to coat different kinds of polymers [32,38]. However, the possible diffusion limitation, i.e., the inhomogeneous formation of conductive phase in the whole matrix, may be a problem in this process. This strategy essentially depends on the diffusion of the polymer monomer or oxidant solution in the preformed hydrogel. Therefore, it is necessary to optimize material selection and process parameters to obtain homogeneous conductive hydrogels through this strategy [39]. Wu et al. prepared a naturally-derived gelatin methyacrylate (GelMA) conductive hydrogel with the user-defined patterns in Figure 3, employing the post polymerization approach [40]. The preformed hydrogels were fully infiltrated in ammonium persulfate (APS) solution as oxidants for subsequent polymerization of aniline monomers. The APS-filled hydrogels were incubated in the aniline solution to form conductive polyaniline (Pani)-GelMA hydrogels. The conductivity of the disk-shaped Pani-GelMA hydrogel with 1.62 mm thickness was confirmed by cyclic voltammetry. Biocompatibility was verified by in vitro cell experiments. The conductive hydrogel had the potential as a new bioelectrical interface for several biomedical applications.

### 2.3. Conductivity Provided by Composite Strategies

This strategy commonly involves addition of conductive particles, including metals, carbon (e.g., C_60_ and graphene), and conductive polymers, into the hydrogel precursor. The conductive particles are homogeneously dispersed in the gelled precursor process, endowing the final hydrogel with conductive properties. The conductivity of the hydrogel can be controlled by adjusting the content of the mixed conductive particles. For example, the conductivity of the chitosan composite hydrogel could increase from 1 × 10^−8^ S/m to 1.33 × 10^−1^ S/m with an increasing amount of conductive chemically converted graphene added in the system as shown in Figure 4A [41]. Compared to the other two strategies, the composite strategy does not require additional steps to remove potential cytotoxic unreactive monomers or oxidants, considering the requirement for biomedical applications, because no polymerization process is required. Therefore, hydrogels prepared by the composite strategy could be directly used in cell experiments or in vitro studies without extra steps to remove impurity [42,43]. Correspondingly, the toxicological evaluations of conductive nanoparticles should not be neglected in biomedical applications [44,45]. Maharjan et al. [46] fabricated gold/silica hybrid (Au/SiO_2_) nanoparticles and incorporated them into a GelMA hydrogel to endow the hydrogel with conductivity (Figure 4B). The charge transfer resistance values of GelMA-Au/SiO_2_ composite hydrogels were higher than those of Au/SiO_2_ nanoparticles measured by electrochemical impedance spectroscopy. Shevach et al. [47] reported and confirmed that the addition of gold nanoparticles to the acellular matrix could give the scaffold conductivity and enhanced myocardial cell adhesion, proliferation and differentiation with extended and arranged morphology.

Another direction of design for making conductive hydrogels through the composite strategy is the formation of hydrogels in electrolyte solutions (e.g., phosphate-buffered saline). This process can directly introduce ions into the hydrogel, thus giving the hydrogel the inherent conductance properties of ions. Especially in biomedical applications, hydrogels are applied in vivo while the body fluid contains various types of ions (i.e., Na^+^, HCO_3_^−^, Ca^2+^, and Cl^−^). The hydrogels infiltrated by body fluids would have corresponding ionic conductance [9,10]. Meanwhile, the addition of ions to the matrix could give extra properties to the hydrogel such as fast gelling [48] and high elasticity [49]. For example, Odent et al. [50] prepared a family of ionic composite hydrogels with excellent mechanical properties, which can be utilized for rapid 3D printing at high resolution. This composite strategy, shown in Figure 5, generates dynamic and reversible Coulombic interaction through the electrostatic interaction of anionic sulfonic groups on the nanoparticles and the cationic quaternary ammonium side groups on the polymer network, so as to enhance the mechanical properties.

## 3. Biomedical Applications of Conductive Hydrogel

A conductive polymer alone is generally brittle, which limits its biomedical applications. However, conductive hydrogels have attracted much attention and are a potential candidate for many biomedical applications because of their unique properties, such as flexibility, conductivity, and biocompatibility [51]. In the following subsections, four types of biomedical applications are described to highlight the versatility of conductive hydrogels.

### 3.1. Biosensors

Biosensors are designed to integrate an electronic transducer and signing devices for motion detection, biological recognition, or electrophysiological signals [7,52,53]. To date, conventional biosensors relying on metallic electrodes are commercially available and highly valued in clinical treatments. However, interfacial impedance and biomechanical mismatches between conventional biosensors and the physiological targets generally occur. Conductive hydrogels have the advantages of easy synthesis, high biocompatibility, tunable strength, and multiresponsive functionality. Compared with conventional biosensors, it is more beneficial to integrate conductive hydrogels with electronic devices for in vivo sensing. Thus, conductive hydrogels as biosensors are of particular interests in the field of bioelectronics [54,55]. Hitherto, biosensors had enormous demands in clinical diagnostics, long-term health monitors and real-time biomolecule sensors [56].

Many wearable biosensors have been recently developed based on conductive hydrogels because of their flexibility and adhesiveness [7,52,57]. Moreover, the correlations between the bioelectrical and mechanical properties have been discussed for better design of wearable biosensors [54,58]. Lu’s group developed a conductive hydrogel based on polydopamine (PDA), graphene oxide, and PAM. This hydrogel was reported to possess self-adhesive ability and was proved to have good biocompatibility both in vitro and in vivo [7]. In another work, a conductive hydrogel composed of PAM, chitosan and polypyrrole was prepared. The hydrogel was utilized as a motion detector, or a stress sensor, and had a tunable conductivity up to 0.3 S/m [59]. Besides, a light-transmitting conductive hydrogel was prepared in situ from hydrophilic PAM with hydrophobic PDA- polypyrrole nanoparticles. The transparency of this hydrogel allowed medical personnel to directly keep track of the patients’ responses while recording biosignals, such as electrocardiograph and magnetocardiography, at the same time [52]. Another conductive hydrogel with high stretchability and reversible adhesiveness consisted of PDA-coated talc nanoflakes and PAM. As a strain sensor, this hydrogel demonstrated high sensitivity with a gauge factor of 0.693 at 1000% strain. Furthermore, the adhesiveness of the hydrogel could be applied to both hydrophilic and hydrophobic materials [57]. Moreover, a 3D printable conductive hydrogel comprising *N*-isopropylacrylamide, multiwalled carbon nanotubes (CNTs) and laponite was developed to provide great possibility for the fabrication of customized wearable biosensors (Figure 6). The latter hydrogel also exhibited high sensitivity towards near-infrared light and temperature, which broadened its biomedical applications [8].

Conductive hydrogels can also be implemented as biosensors to detect biomolecules or human metabolites. The biosensors are fabricated using specifically modified hydrogels to respond to target analytes. Upon interacting with the target analytes, the hydrogels tend to undergo apparent and rapid physical changes for easy observation [60]. For example, a biosensor platform based on a polyaniline hydrogel modified with platinum nanoparticles was developed, which can be utilized to sense human metabolites such as uric acid, cholesterol and triglycerides (Figure 7) [61]. Furthermore, a lactate biosensor was prepared on the basis of a dimethylferrocene-modified linear poly(ethylenimine) hydrogel. This conductive hydrogel operated as a self-powered biosensor when coupled with a lactate biocathode [62]. In another work, a methacrylated collagen hydrogel modified with polypyrrole showed conductivity and injectability and was capable of monitoring glucose levels on an electrode surface or in a piece of porcine meat [63]. Moreover, a biosensor was obtained by dripping a solution containing electrochemiluminescent luminophore functionalized silver nanoparticles on a polyaniline-phytic acid-conductive hydrogel. This composite hydrogel was able to perform in situ monitoring of hydrogen peroxide provided by cells and was anticipated to detect other biomolecules [64].

To sum up, conductive hydrogels with diverse functionality, adjustable strength and a cytofriendly niche could be designed as wearable motion sensors or biomolecules detectors. It is highly possible to envision precise movement being recorded, or versatile biomolecules being sensed, by conductive hydrogels to facilitate clinical diagnostics.

### 3.2. Wound Dressing

Skin covers the entire body, regulates physiological temperature and senses external stimuli. Chronic skin wounds have become a crucial issue because of their high morbidity leading to many potential complications [51,65]. Conductive materials were reported to upregulate cellular activities of fibroblasts [66]. Conductive hydrogels as wound dressing materials have several merits, such as avoidance of microbial infection, maintenance of moisture environment and promotion of hemostasis and adhesiveness [65,67]. Guo et al. [66] reported a conductive hydrogel prepared by mixing quaternized chitosan-g-polyaniline and benzaldehyde group functionalized poly(ethylene glycol)-co-poly(glycerol sebacate) solutions. This injectable conductive hydrogel exhibited self-healing ability, blood clotting capacity, antibacterial activity and free radical scavenging capacity. In addition, the results obtained from a full-thickness skin defect model and histopathologic examination suggested that the hydrogel could be implemented as a wound dressing. Another conductive hydrogel was obtained from the copolymerization of acrylamide and acrylic acid in PDA-decorated CNTs dispersion. This hydrogel showed high tissue adhesiveness, long-lasting moisture capacity and wide-range thermal tolerance, and, as a wound dressing, the hydrogel protected skin from frostbites or burns [27]. Furthermore, a conductive hydrogel based on *N*-carboxyethyl chitosan and an oxidized hyaluronic acid-graft-aniline tetramer was developed as a wound dressing. The evaluation in a full-thickness skin defect model indicated that the conductive hydrogel could facilitate wound healing by growing thick granulation tissue, upregulating collagen disposition and promoting more angiogenesis [28]. Wang et al. [68] developed a conductive hydrogel by polymerization of poly (2-hydroxyethyl methacrylate), polypyrrole and 3-sulfopropyl methacrylate. The hydrogel demonstrated low protein absorption and could prevent damage to newly-formed tissues when the dressing was replaced. Additionally, a diabetic rat model demonstrated that this conducive hydrogel could combine with electrical stimulation to treat chronic wounds. In another work, a conductive composite hydrogel based on hyaluronic acid-graft-dopamine and reduced graphene oxide (rGO) was prepared. A conventional antimicrobial drug, such as doxycycline, could be encapsulated in the composite hydrogel and the photothermal property of rGO further boosted the antibacterial activity in vivo [69]. Besides, a conductive hydrogel was synthesized via a supramolecular assembly of PDA decorated silver nanoparticles (PDA@Ag NPs), polyaniline and polyvinyl alcohol. This hydrogel demonstrated a therapeutic effect on diabetic foot wounds. It also functioned as epidermal sensors to record the healing process, as shown in Figure 8 [70]. To reduce infection in wound healing, a conductive hydrogel based on chitosan, gelatin-grafted-dopamine and PDA coated CNTs was developed. The hydrogel was able to carry extra antimicrobial agents, and its photothermal property associated with the CNT produced better antibacterial associated with a CNT effect in an infected full-thickness mouse skin defect wound [67]. Recently, a rapid-forming (within 1 min) conductive hydrogel consisting of poly (3,4-ethylenedioxythiophene), poly (styrenesulfonate) and guar slime was reacted. The wound healing ability of the conductive hydrogel was evaluated by a skin wound model on the nape of the rats, which undergoes rapid and frequent movements. Data for in vivo experiment showed that the conductive hydrogel was able to accelerate wound healing, promote epithelialization and facilitate hair follicle regeneration [71].

In conclusion, antibacterial activity, tissue adhesiveness, hemostatic ability, conductivity and biocompatibility are crucial criteria for designing wound dressing. Conductive hydrogels have the advantage of convenient preparation with multifunctionality. It is of interest to incorporate all the above-mentioned features to develop clinical-ready wound dressings and artificial skin.

### 3.3. Drug Delivery

Many smart hydrogels with stimulus-responsive abilities, such as those responding to ultrasound, light and magnetic signals, have been developed as vehicles for drug delivery. These stimulus-responsive hydrogels are generally triggered by specific or large-sized equipment. In contrast, conductive hydrogels are activated by electric signals that are easier to generate and manipulate [53,72]. With such a convenient feature, conductive hydrogels are considered as powerful candidates for smart and precise drug delivery devices. For instance, a dual-stimuli responsive composite hydrogel based on polypyrrole nanoparticles and PLGA-PEG-PLGA hydrogel was reported, as shown in Figure 9. Fluorescein as a model drug was encapsulated in the conductive hydrogel to investigate the correlation between the drug release profile and external electric current. A possible mechanism of electric-field-triggered drug release was proposed for the system, which paved the way for the development of novel electric-responsive drug release vehicles [72]. In another example, a conductive hydrogel for drug release was prepared from the hybrid of acrylic acid, *N*,*N*-methylenebisacrylamide and graphene. In addition to its conductive property, this hydrogel showed pH sensitivity and offered a rubber-like mechanical behavior. A pulsatile drug release profile could be obtained when an electric field was implemented [73]. Another pH sensitive conductive hydrogel was composed of chitosan-graft-polyaniline and oxidized dextran, which also exhibited good injectability and antibacterial ability. The drug release profiles of two model drugs, amoxicillin and ibuprofen, could be adjusted by varying pH or applying an electric field. Moreover, cytotoxicity tests and in vivo experiment confirmed the biocompatibility of the conductive hydrogel. The unique features of the hydrogel suggested the possibility for a powerful drug delivery carrier with precise dosage and location control [29]. In another work, a conductive hydrogel with the potential of being an on-demand drug release vehicle, was achieved by reacting dextran and an aniline trimer. The on-off drug release behavior of the hydrogel was confirmed by applying different voltages, that is, a larger amount of drug was released with higher voltages and vice versa. Date from in vitro and in vivo experiments showed that the hydrogel possessed good biocompatibility. The concept of the on-off drug release provided better opportunity of the conductive hydrogel for clinical uses [30]. Furthermore, a conductive hydrogel consisting of chitosan–aniline oligomer and agarose was prepared with a tunable swelling ratio from ~800% to ~300% and an adjustable degradation rate. The aniline motif in the hydrogel played a key role for on-demand drug release ability and provided a proper conductivity (~10^−4^ S/cm) for promoting cell proliferation [74]. In addition, a conductive hydrogel based on poly(dimethylacrylamideco-4-methacryloyloxy benzophenone-co-4-styrenesulfonate and poly(3,4-ethylenedioxythiophene) as a drug delivery system was developed, and two model drugs, fluorescein and dexamethasone, were encapsulated in the hydrogel to evaluate the drug release function. The hydrogel was coated on an electrode to fabricate a reusable system, which allowed direct delivery of the drug to a neural interface periodically [75].

To summarize, drug delivery systems designed from conductive hydrogels demonstrate release profiles with dosage and spatial control. Sequential and long-term (release profile up to months) drug release based on conductive hydrogels are worthy of investigation. Encapsulating more therapeutic drug molecules, antibacterial agents, genes or living cells in the conductive hydrogels may be the future targets for new types of drug delivery vehicles.

### 3.4. Tissue Regeneration

Conductive hydrogels as matrices are able to mimic the extracellular matrix as well as transmit electrical signals to excitable cells such as muscle cells, neuronal cells and myocardial cells [53,65]. Conductive hydrogels are able to regulate the proliferation and differentiation of these excitable cells through further electrical stimulation [59,65]. For example, an engineered cardiac patch was prepared from a composite conductive hydrogel incorporating CNTs and GelMA. Neonatal rat cardiomyocytes seeded on this conductive hydrogel demonstrated spontaneous synchronous beating and a lower excitation threshold compared to those seeded on the nonconductive hydrogel, suggesting that this hybrid conductive hydrogel may have potential in cardiac tissue engineering [76]. Another conductive hydrogel, composed of a rGO sheet and PAM, was prepared for skeletal muscle engineering. This composite conductive hydrogel displayed a muscle tissue-like stiffness (50 kPa), where the embedded mouse myoblastic C2C12 cells were able to proliferate and express a higher level of myogenic-associated genes [77]. Besides, a conductive hydrogel based on chitosan, PDA, and graphene oxide (GO) was prepared. The hydrogel had a conductivity range from 0.57 mS/cm to 1.22 mS/cm as well as self-healing property and adhesiveness, which offered a cytofriendly niche for human embryonic stem cell-derived fibroblasts to proliferate and for cardiomyocytes to beat [78]. Besides, a conductive hydrogel based on chitosan, PDA and GO was prepared. The hydrogel had a conductivity range from 0.57 mS/cm to 1.22 mS/cm as well as self-healing properties and adhesiveness, which offered a cytofriendly niche for human embryonic stem cell-derived fibroblasts to proliferate and for cardiomyocytes to beat [23]. Furthermore, a 3D printable conductive hydrogel was prepared from poly(3,4-ethylenedioxythiophene), polystyrene sulfonate (PSS) and polyethylene glycol diacrylate. After being printed into different constructs, the hydrogel was integrated with a GelMA precursor that contained a dorsal root ganglion cell (DRG cell) suspension through photocrosslinking. DRG cells on the conductive constructs were able to proliferate, while their neuronal differentiation was upregulated via electrical stimulation [23]. Moreover, Guo et al. reported an injectable conductive hydrogel based on aniline tetramer grafted dextran and *N*-carboxyethyl chitosan. This conductive hydrogel showed good cytocompatibility in an in vitro experiment and, most interestingly, demonstrated a therapeutic effect on a rat skeletal muscle injury model in vivo, as illustrated in Figure 10 [31]. Recently, a composite conductive hydrogel comprising amino-terminated 4-armed polyethylene glycol, diacerein and a GO nanosheet was developed. The conductive hydrogel had good injectability and demonstrated a therapeutic effect on a rat spinal cord injury model owing to conductivity from the GO nanosheet and an anti-inflammatory response from diacerein [79].

According to the above-mentioned literature, conductive hydrogels offer a biocompatible environment and provide a platform for delivery of electrically excitable cells. Although these references reveal many promising in vitro results, some hurdles remain to be overcome for animal and clinical transitional studies. Long-term stability of mechanical, electrical and biological properties of the conductive hydrogels are primary foci. Moreover, combination with convenient processability and precise microfabrication would allow a more sophisticated design of conductive hydrogels as biomimetic constructs. Last, but not least, the cell-matrix interaction and the in vivo performance of conductive hydrogels require more interdisciplinary collaboration to truly achieve the goal for tissue engineering.

## 4. Summary and Perspective

The development of conductive hydrogels with tunable physical and biochemical properties is a promising research area in the biomedical field, but most of the biomedical applications based on conductive hydrogels are still in their infancy. To the best of our knowledge, conductive hydrogels have only been studied for their therapeutic effects in rat models within the recent five years. The long-term stability, functionality and cytocompatibility of conductive hydrogels deserve more investigation before conductive hydrogels can be implemented in larger animal models or clinical studies. Therefore, it is of particular importance to understand the current design strategies of conductive hydrogels and seek improvement. Three different strategies, including the conductivity provided by in situ polymerization of conductive polymers, post polymerization of conductive polymers and composite strategies, have been widely used to design conductive hydrogels. Conductive hydrogels designed through these three strategies have been extensively studied for possible uses in the biomedical field because of their electroactive properties that contribute to enhance cellular behaviors and biocompatibility. In this review, the focus has been laid on the classification and differences among the three design strategies, and applications of conductive hydrogels in the biomedical area, including biosensors, wound healing, drug delivery and tissue regeneration. Among the design strategies, the in situ polymerization strategy is a one-step preparation to avoid complex polymerization process and the post polymerization strategy is an alternative coating tool to endow an already formed hydrogel with conductivity. The composite strategy avoids the presence of unreacted monomers or oxidants with potential cytotoxicity, but the toxicology of conductive nanoparticles and ions cannot be ignored.

Selection of different conductive strategies and various smart materials systems is the key to designing conductive hydrogels that have multiple functions (e.g., pH, strain and thermo-responsiveness). A mostly studied application of conductive hydrogels in the biomedical field is the biosensor, including wearable motion sensors and implantable biomolecule detectors. Meanwhile, these two key factors are also important for other applications such as wound healing, tissue regeneration, and as drug delivery where stimuli-responsive conductive hydrogels serve as on-off devices to control the release of drugs. However, in addition to meeting the magnitude requirement of conductivity for these conductive biomaterials, some major obstacles rest with resolving problems related to their processability and cytotoxicity. 

Currently, many researchers combine microfabrication technology of conductive hydrogels with the current trend of manufacturing, which can accurately reproduce the structural characteristics of the natural extracellular matrix and enhance the function of a conductive hydrogel in a physiological environment. Particularly, the combination of 3D bioprinting, i.e., cell-containing printing technology, with biocompatible conductive hydrogels has great potential in the tissue engineering of nerve and myocardium. At the same time, using one material to achieve various functions is a future trend for conductive hydrogels. Most of the reported conductive hydrogels do not have multiple biological functions, such as having functions in both biosensing and tissue regeneration. Integrating multiple biological functions into one hydrogel will be of great significance. The progress of conductive hydrogels for biomedical applications, in our opinion, will be largely dependent on the balanced development of conductive strategies, material choices, versatile functionality and various fabrication technologies. The conductive hydrogel prepared by weighing these key factors will be conducive to the future development of biomedical applications.

## Figures and Tables

**Figure 1 molecules-25-05296-f001:**
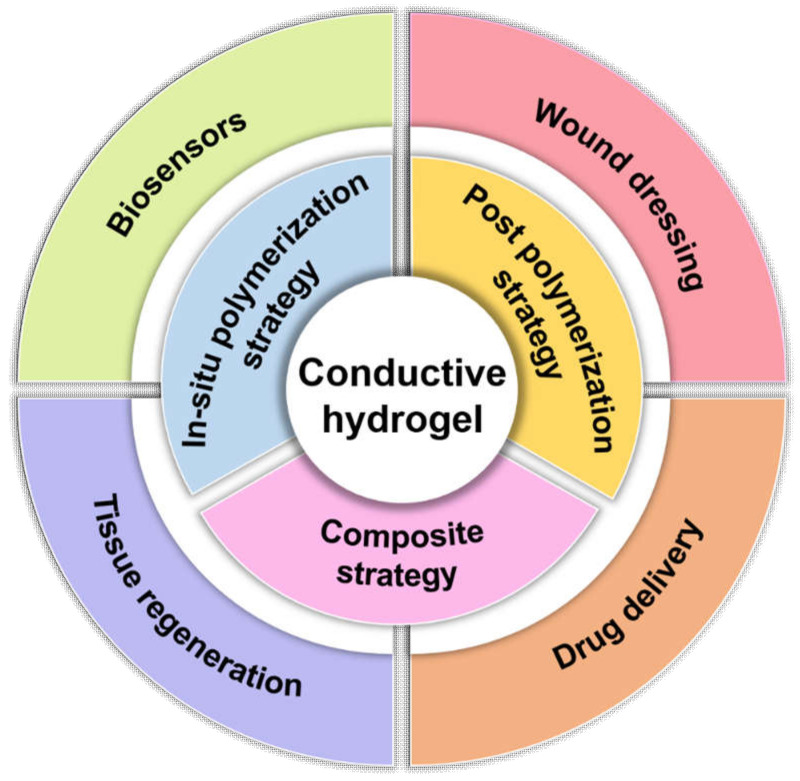
Summary for the design strategies and biomedical applications of conductive hydrogels.

**Figure 2 molecules-25-05296-f002:**
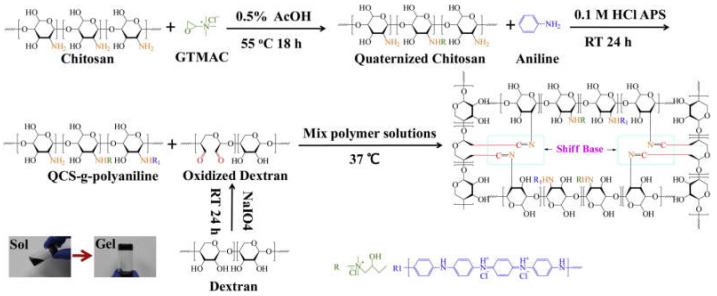
Scheme for the formation of quaternized chitosan (QCS)–polyaniline/oxidized dextran hydrogel. GTMAC is the abbreviation of glycidyltrimethylammonium chloride [18]. Reprinted from [18] with permission from Elsevier.

**Figure 3 molecules-25-05296-f003:**
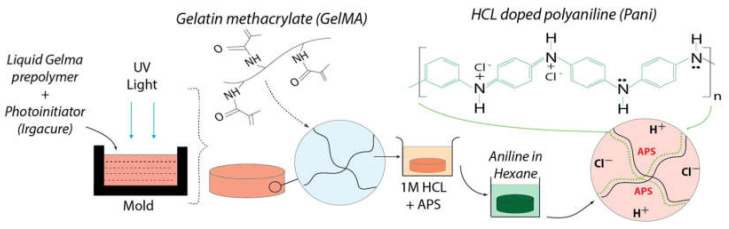
Fabrication of Pani-GelMA conductive hydrogels by a post polymerization strategy through immersion in preformed hydrogel in an oxidant solution followed by subsequent immersion in an aniline/hexane solution for obtaining the conductive hydrogel [40]. Reprinted from [40] with permission from Elsevier.

**Figure 4 molecules-25-05296-f004:**
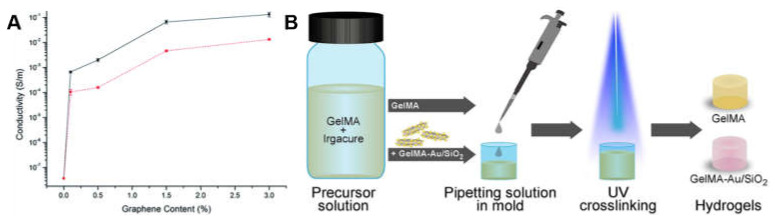
(**A**) Conductivity changes based on the graphene content of composite hydrogels produced using lactic acid (black) and acetic acid (red) [41]. Reprinted from [41] with permission from Royal Society of Chemistry. (**B**) A schematic showing the procedure for preparing composite conductive hydrogel [46]. Reprinted from [46] with permission from Elsevier.

**Figure 5 molecules-25-05296-f005:**
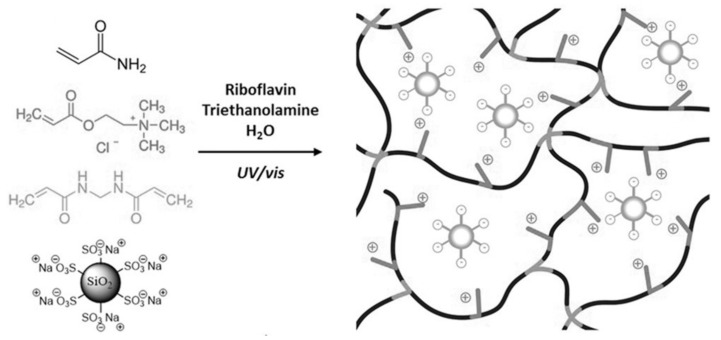
Schematics presenting the formation of the ionic composite hydrogels via photopolymerization with the presence of sulfonate modified SiO_2_ nanoparticles [50]. Reprinted from [50] with permission from WILEY-VCH.

**Figure 6 molecules-25-05296-f006:**
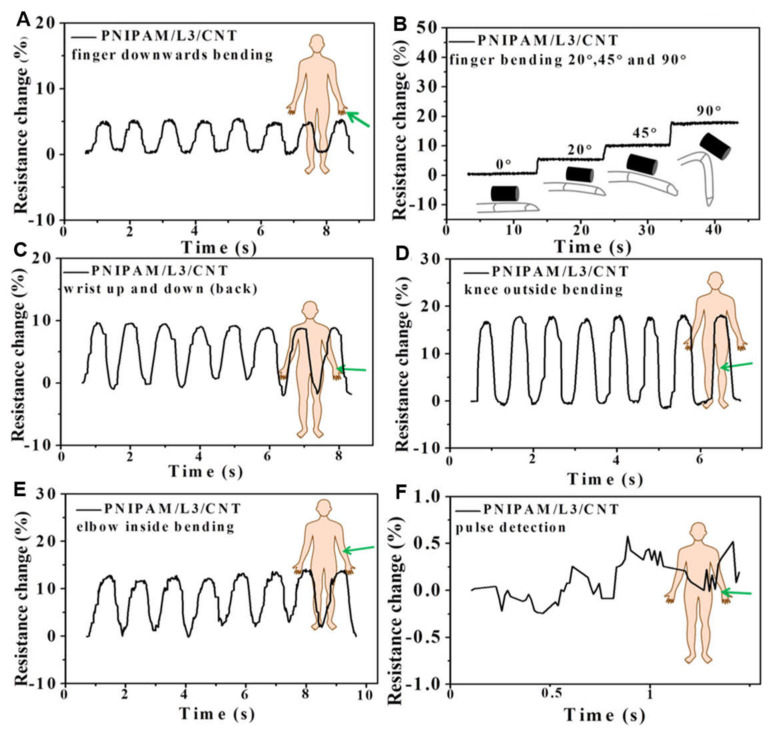
Relative resistance changes for *N*-isopropylacrylamide (PNIPAM)/laponite (L)/carbon nanotube (CNT) hydrogels of (**A**,**B**) index finger two direction bending, (**C**) wrist motions, (**D**) knee bending, (**E**) elbow bending, and (**F**) pulse detection [8]. Reprinted from [8] with permission from American Chemical Society.

**Figure 7 molecules-25-05296-f007:**
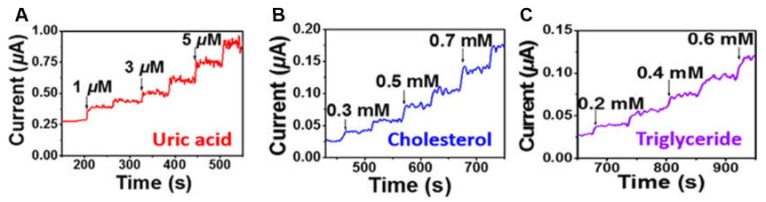
The conductive hydrogel as a biomolecular detector for different concentrations of (**A**) uric acid, (**B**) cholesterol, and (**C**) triglyceride [61]. Reprinted from [61] with permission from American Chemical Society.

**Figure 8 molecules-25-05296-f008:**
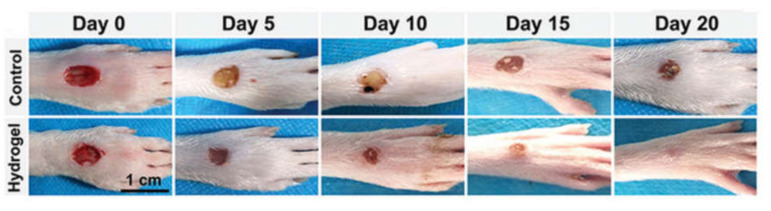
Representative photos of the diabetic foot in rats after being treated with the control group (PBS) or the conductive hydrogel containing PDA@Ag NPs [70]. Reprinted from [70] with permission from WILEY-VCH.

**Figure 9 molecules-25-05296-f009:**
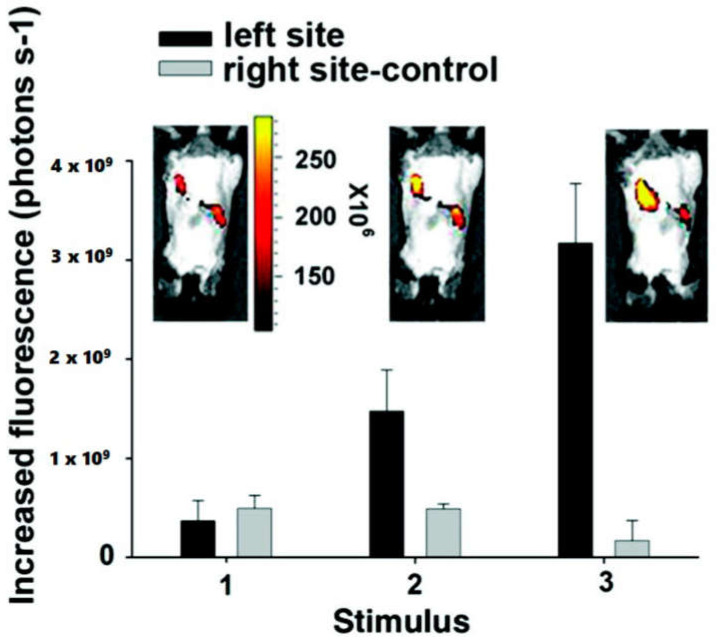
Fluorescent images of the implanted conductive hydrogel in vivo under an electric field of −1.5 V/cm. The labels on the *X*-axis represent (1) before applying electric field and (2,3) after applying electric field on the left site of the imaged animals. The right side of each image is the control side without applying any electric field [72]. Reprinted from [72] with permission from American Chemical Society.

**Figure 10 molecules-25-05296-f010:**
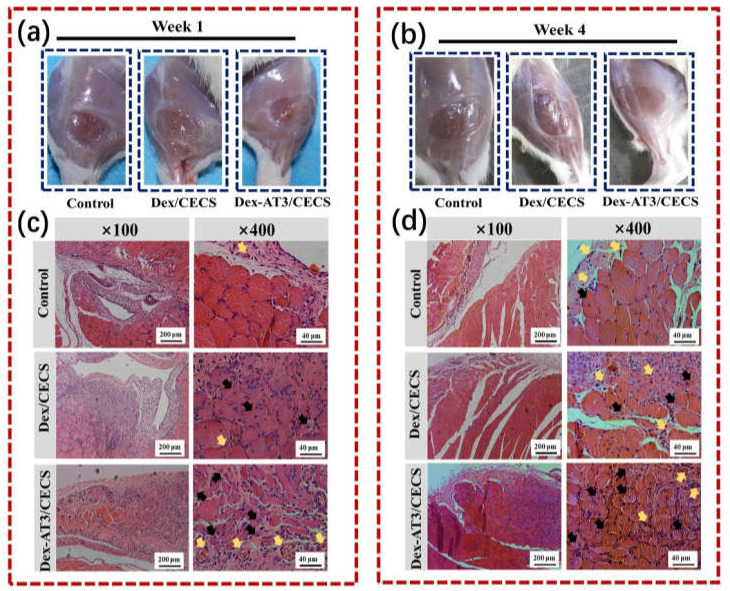
In vivo evaluation of skeletal muscle tissue regeneration in a muscle loss injury rat model. The morphology of tibialis anterior muscle after injury and treatment for 1 week (**a**) and 4 weeks (**b**); hematoxylin and eosin staining images of hydrogels and PBS control after implantation for 1 week (**c**) and 4 weeks (**d**). Black arrows represent centronucleated myofibers and yellow arrows represent newly formed blood vessels. Reprinted with permission from [31]. Reprinted from [31] with permission from Elsevier.

**Table 1 molecules-25-05296-t001:** Advantages and disadvantages of different design strategies for preparing conductive hydrogels.

Design Strategies	Advantages	Disadvantages
In situ polymerization	One-step preparation Homogenous polymerization	Potential cytotoxic unreactive oxidants and monomers Requirement of designing chemical synthesis process
Post polymerization	Endowing conductivity to preformed hydrogel Potential conductive coating method	Potential cytotoxic unreactive oxidants and monomers Extra polymerization step
Composite strategies	Controllable conductivity No potential cytotoxic unreactive oxidants and monomers	Heterogeneous distribution of additives Toxicity of conductive additives

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
