# Peer review of "Design Strategies of Conductive Hydrogel for Biomedical Applications"

_molecules, 2020, doi:10.3390/molecules25225296_

Round 1

Reviewer 1 Report

The authors conducted a review about the design strategies of conductive hydrogel for biomedical applications. Firstly, they summarized the different strategies for the construction of conductive hyrogel. Secondly, they reviewed the applications of conductive hyrogel in the fields of biosensors, tissue regeneration, wound dressing and drug delivery. Finally, conclusion and perspective are listed at the end of the paper. Some suggestions are listed below.

1) Figure 1, it is nice but I suggest to change the caption into: "summary for the design strategies and biomedical applications of conductive hydrogels".

2) Table to summarize the advantages and disadvantages of different design strategies of conductive hydrogel should be added.

3) Perspective part is simple and superficial.

Author Response

Reviewer 1:

The authors conducted a review about the design strategies of conductive hydrogel for biomedical applications. Firstly, they summarized the different strategies for the construction of conductive hyrogel. Secondly, they reviewed the applications of conductive hyrogel in the fields of biosensors, tissue regeneration, wound dressing and drug delivery. Finally, conclusion and perspective are listed at the end of the paper. Some suggestions are listed below.

1) Figure 1, it is nice but I suggest to change the caption into: "summary for the design strategies and biomedical applications of conductive hydrogels".

Ans. Thank you for your kind suggestion. We have already modified the figure caption of Figure 1 according to your suggestion.

2) Table to summarize the advantages and disadvantages of different design strategies of conductive hydrogel should be added.

Ans. Thank you for your kind suggestion. We have already supplemented one table including the pros and cons of different design strategies of conductive hydrogel on Page 3.

3) Perspective part is simple and superficial.

Ans. Thank you for your kind suggestion. A modified version of the Summary and Perspective section was added in the manuscript on Page 13 Lines 389 to Line 395. We stated the versatility of conductive hydrogels, mentioned the current development and need of conductive hydrogels, and emphasized the reason why the understanding of designing strategies of the conductive hydrogel is crucial. “The development of conductive hydrogels with tunable physical and biochemical properties is a promising research area in the biomedical field, but most of the biomedical applications based on conductive hydrogels are still in their infancy. To the best of our knowledge, conductive hydrogels have only been studied for their therapeutic effects in rat models within the recent five years. The long-term stability, functionality, cytocompatibility of conductive hydrogels deserve more investigations before conductive hydrogels could be implemented in larger animal models or clinical study. Therefore, it is of particular importance to understand the current designing strategies of conductive hydrogels and seek for improvement.”

Reviewer 2 Report

The article is an interesting review of the development and use of conductive hydrogels in the biomedical field. Three main approaches for hydrogel treatment are considered (in-situ polymerization, post-polymerization, composite strategies) and four significant applications are described in detail (biosensors, wound dressing, drug delivery, tissue engineering). There are a few questions that the Authors should successively resolve before the manuscript may be considered suitable for publication, plus some minor adjustments in the text. All issues are listed below.

Question 1(about section 2.3): does the spatial distribution of conductive particles within the medium play any role in determining the overall conductivity of the hydrogel? The same question applies to the case of ion-conductive hydrogels.considered at the end of page 5.

Question 2: does the ion-dependent conductivity of hydrogels depend on an applied electric field? If so, is there a way to characterize an optimal range of values for such an applied field, or does it strongly depend on the specific configuration?

Question 3 (section 3.2): what is (are) the basic physical mechanism(s) which allow a conductive hydrogel to perform wound healing? Providing such information would improve the clarity of the presentation.

Question 4 (section 3.3): this is similar to question 2.

Question 5 (section 3.3): are hydrogel-based drug delivery carriers designed to travel safely throughout human blood without giving rise to toxic reactions by the body system?

Question 6 (section 3.4): is there a specific waveform pattern and/or intensity for the stimulating electric signal in order to drive a specific cellular differentiation (neurional, for example, cf. lines 357-359)?

MINOR ISSUES:

line 14: biomedical fields -----> biomedical field

line 16: suit the different -----> suit different

line 17: as scaffolds with -----> as a scaffold with

line 20: developed for applications in biomedical fields -----> development for applications in the biomedical field

line 58: tissue regeneration [31][ref].: what does [ref] refer to???

line 116: measured by a two-point probe measurement of 165.5 kΩ.

The reported value is a resistance. It is not clear what component such resistance refers to.

line 126: particles are the homogeneously -----> particles are homogeneously

line 235: covers entire ----> covers the entire

line 373: According to the above literature, -----> According to the above mentioned literature,

line 374: Although these literatures -----> Although these refernces

Author Response

Reviewer 2:

The article is an interesting review of the development and use of conductive hydrogels in the biomedical field. Three main approaches for hydrogel treatment are considered (in-situ polymerization, post-polymerization, composite strategies) and four significant applications are described in detail (biosensors, wound dressing, drug delivery, tissue engineering). There are a few questions that the Authors should successively resolve before the manuscript may be considered suitable for publication, plus some minor adjustments in the text. All issues are listed below.

Question 1(about section 2.3): does the spatial distribution of conductive particles within the medium play any role in determining the overall conductivity of the hydrogel? The same question applies to the case of ion-conductive hydrogels.considered at the end of page 5.

Ans. Thank you for your kind suggestion. All research papers on hydrogels containing conductive particles need to address the problem of dispersion. At present, the dispersibility of the composite conductive hydrogel reported is homogeneous, although the material system varies greatly. Meanwhile, cell culture medium exists in two states in the hydrogel system, inside and surrounding the hydrogel. Conductivity, as an intrinsic property of the hydrogel, is affected by the presence of ions in the cell culture medium. However, the ions in the environment can only affect the hydrogel gradually, and cannot determine the inherent properties of the hydrogel. The ion-conductive hydrogel not only is simply mixed with ions, but also plays a role in the structure to facilitate the formation of crosslinking networks. Therefore, the relationship between ionic distribution and conductivity in an ion-conductive hydrogel is the first concern that a research team needs to address.

Question 2: does the ion-dependent conductivity of hydrogels depend on an applied electric field? If so, is there a way to characterize an optimal range of values for such an applied field, or does it strongly depend on the specific configuration?

Ans. Thank you for your kind suggestion. The conductivity of the conductive hydrogel, regardless of the preparation strategy, is the intrinsic property that cannot be simply changed by applying an external electric field of different intensity. At present, there are many methods and conditions to obtain the conductivity. However, due to the presence of water in the hydrogel, the electrochemical properties of water need to be considered, such as the electrochemical window value of water.

Question 3 (section 3.2): what is (are) the basic physical mechanism(s) which allow a conductive hydrogel to perform wound healing? Providing such information would improve the clarity of the presentation.

Ans. The authors are grateful for the suggestion. To address this suggestion, a new sentence “Conductive materials were reported to upregulate cellular activities of fibroblasts [66]” was added at line 241-242 to explain why conductive hydrogel can facilitate wound healing. Skin is mostly composed of fibroblasts and these cells are electrically sensitive. Therefore, promoting the cellular activities of fibroblasts can accelerate wound healing.

References:

[66] Zhao, X., Wu, H., Guo, B., Dong, R., Qiu, Y., & Ma, P. X. (2017). Antibacterial anti-oxidant electroactive injectable hydrogel as self-healing wound dressing with hemostasis and adhesiveness for cutaneous wound healing. Biomaterials, 122, 34-47.

Question 4 (section 3.3): this is similar to question 2.

Ans. Thank you for your kind suggestion. The conductivity of the conductive hydrogel, regardless of the preparation strategy, is the intrinsic property that cannot be simply changed by applying an external electric field of different intensity. At present, there are many methods and conditions to obtain conductivity. However, due to the presence of water in the hydrogel, the electrochemical properties of water need to be considered, such as the electrochemical window value of water.

Question 5 (section 3.3): are hydrogel-based drug delivery carriers designed to travel safely throughout human blood without giving rise to toxic reactions by the body system?

Ans. The authors are thankful that the reviewer rose this concern. As far as we know, the researches cited in this review have only gone through animal experiments. There is no clinical study or trial reported so far.

Question 6 (section 3.4): is there a specific waveform pattern and/or intensity for the stimulating electric signal in order to drive a specific cellular differentiation (neurional, for example, cf. lines 357-359)?

Ans. Thank you for the interesting question. To the best of our knowledge, a direct electric current should be implemented when stimulating specific cells, and the intensity of the electric stimulation should match the innate electric conductivity of the specific cells, such as native myocardium (1-4 mS/cm) [1], proliferating neurons (1–10 S/m) [2], and native human skeletal muscle (4.5×10-2 – 8×10-3 mS/cm) [3].

References:

  1. Jing, X.; Mi, H.-Y.; Napiwocki, B. N.; Peng, X.-F.; Turng, L.-S., Mussel-inspired electroactive chitosan/graphene oxide composite hydrogel with rapid self-healing and recovery behavior for tissue engineering. Carbon 2017, 125, 557-570.
  2. Zhang, K.; Li, J.; Jin, J.; Dong, J.; Li, L.; Xue, B.; Wang, W.; Jiang, Q.; Cao, Y., Injectable, anti-inflammatory and conductive hydrogels based on graphene oxide and diacerein-terminated four-armed polyethylene glycol for spinal cord injury repair. Materials & Design 2020, 196, 109092.
  3. Guo, B.; Qu, J.; Zhao, X.; Zhang, M., Degradable conductive self-healing hydrogels based on dextran-graft-tetraaniline and N-carboxyethyl chitosan as injectable carriers for myoblast cell therapy and muscle regeneration. Acta Biomaterialia 2019, 84, 180-193.

MINOR ISSUES:

line 14: biomedical fields -----> biomedical field

Ans. Thank you for your kind suggestion. We have already corrected this part according to your advice.

line 16: suit the different -----> suit different

Ans. Thank you for your kind suggestion. We have already corrected this part according to your advice.

line 17: as scaffolds with -----> as a scaffold with

Ans. Thank you for your kind suggestion. We have already corrected this part according to your advice.

line 20: developed for applications in biomedical fields -----> development for applications in the biomedical field

Ans. Thank you for your kind suggestion. We have already corrected this part according to your advice.

line 58: tissue regeneration [31][ref].: what does [ref] refer to???

Ans. Thank you for your kind suggestion. We have already modified the manuscript.

line 116: measured by a two-point probe measurement of 165.5 kΩ.

The reported value is a resistance. It is not clear what component such resistance refers to.

Ans. Thank you for your kind suggestion. We have already removed this sentence from our manuscript because it hinders the understanding of the manuscript.

line 126: particles are the homogeneously -----> particles are homogeneously

Ans. Thank you for your kind suggestion. We have already corrected this part according to your advice.

line 235: covers entire ----> covers the entire

Ans. Thank you for your kind suggestion. We have already corrected this part according to your advice.

line 373: According to the above literature, -----> According to the above mentioned literature,

Ans. Thank you for your kind suggestion. We have already modified the description here (change “According to the above literature,” to “According to the above-mentioned literature,”).

line 374: Although these literatures -----> Although these refernces

Ans. Thank you for your kind suggestion. We have already modified the description here (change “Although these literatures” to “Although these references”).

Round 2

Reviewer 1 Report

The authors have clarified my concerns.